# Generating Rough Stereoscopic 3D Line Drawings from 3D Images

Lesley Istead*     Andreea Pocol†     Craig S. Kaplan‡     Isaac Watt §     Nick Lemoing ¶     Alicia Yang ‖

University of Waterloo

## ABSTRACT

We present a method to produce stylized drawings from stereoscopic 3D (S3D) images. Taking advantage of the information provided by the disparity map, we extract object contours and determine their visibility. The discovered contours are stylized and warped to produce an S3D line drawing. Since the produced line drawing can be ambiguous in shape, we add stylized shading to provide monocular depth cues. We investigate using both consistently rendered shading and inconsistently rendered shading in order to determine the importance of lines and shading to depth perception.

**Keywords:** Stereoscopic 3D, non-photorealistic rendering.

**Index Terms:** Computing methodologies—Computer graphics—Rendering—Non-photorealistic rendering; Computing methodologies—Computer graphics—Image manipulation—Image processing; Computing methodologies—Computer graphics—Image manipulation—Image-based rendering

## 1 INTRODUCTION

Stereoscopic 3D is used in a variety of art forms, such as photography and film, to create the effect of depth. The perceived depth can provide a greater sense of reality, create an immersive or engaging experience, and serve as an artistic medium to induce emotional responses in the viewer. S3D creates this sense of depth by presenting a slightly different image to each eye. The left and right images exhibit horizontal separation between objects, which is interpreted as depth by the brain. Producing S3D content is challenging and emphasis must be placed on **consistency**, ensuring that the object(s)/scene visible in both views match(es) exactly to produce a comfortable viewing experience and correct depiction of depth [14, 20].

Line, or pen-and-ink, drawings are one of the oldest S3D art forms, dating back to Sir Charles Wheatstone's original drawings in the 1830s [29]. This format persists today in comics and diagrams. Although S3D line drawings can be produced from 3D meshes using automated algorithms, producing S3D line drawings from S3D photos has not received significant attention.

One possibility for producing line drawings from S3D photos is to use a stereoscopic 3D stylization algorithm such as the layer-based method presented by Northam et al. [19, 20]. This approach divides the S3D image and disparity into layers by disparity, such that each layer only contains pixels from a single disparity, then applies stylization to these layers. Note that disparity is inversely proportional to depth and conveys the horizontal separation between points in the left and right views. While their approach can be used

---

*e-mail: lanortha@uwaterloo.ca
†e-mail: apocol@uwaterloo.ca
‡e-mail: csk@uwaterloo.ca
§e-mail: isaac.watt@uwaterloo.ca
¶e-mail: nlemoing@uwaterloo.ca
‖e-mail: r39yang@uwaterloo.ca

with a variety of artistic styles and filters, contours and line drawing were not considered.

If we try to produce a line drawing using this method, the results are displeasing. This is because object contours will be conflated with other contours, such as lighting and texture boundaries. Thus, the final result contains lines that do not convey shape. We could use the additional information provided by a disparity map to isolate object contours. However, the layer-based approach cannot be used to extract these contours from the disparity map, because layers contain pixels with the same disparity value. Figure 1 illustrates this issue. Note how the contours found for the disparity layer do not correspond to object contours. Instead, they correspond to the contours of the region with the given disparity.

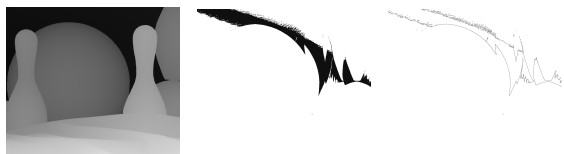

(a) Left disparity map  (b) A disparity layer mask; all pixels in black have the same disparity  (c) Contours found in the disparity layer

Figure 1: Line drawing from disparity layers. Note how each disparity layer only contains pixels of one value. Hence, extracting lines from such layers produces the edges of the layer pixels instead of the desired object contours.

In this paper, we present a method to produce stylized stereoscopic 3D line drawings from the disparity maps of 3D photos using stereoscopic warping instead of layers. While constructing this method, we observed that some drawings of simple objects were ambiguous and did not uniquely identify the 3D shape. For example, the contour of a sphere is a circle and could represent either a flat circle or a sphere. Shading, a monocular depth cue, can help resolve these ambiguities. Hence, we also investigate the effects of adding stylized shading to our produced drawings.

The line extraction and rendering algorithm is presented in Section 3, and our shading method is discussed in Section 4. Our results are presented in Section 5. Finally, we present an evaluation of our results to verify their 3D comfort and depth quality in Section 6.

## 2 BACKGROUND

A line drawing is a simplistic representation of an object or scene. It is comprised entirely of lines, which may be stylized, and which do not contain shading or colour. Despite the simplicity of such drawings, these drawings are capable of accurately conveying the subject that they depict. Hertzmann indicates line drawings "work" because they "approximate realistic renderings" [10].

Where do artists draw lines? A line drawing study by Cole et al. examined where artists draw lines for a variety of objects [6]. They observed that contours, creases and folds – which describe the shape of the object – were drawn, but lines depicting shadows or

highlights were not. This was also observed by Hertzmann et al. while rendering line drawings for smooth meshes [11].

In a stereoscopic 3D line drawing, contours, creases and folds give the primary sense of an object's 3D shape and depth. Without other S3D cues from which to infer depth, it is important that these lines are as consistent as possible between left and right views. Inconsistencies can cause viewing discomfort from binocular rivalry, a phenomenon where the brain rapidly switches between left and right eyes because the images differ, as well as double vision, detracting from the viewer's perception of object depth. Previous studies have shown that these S3D lines alone sufficiently convey object shape and depth for many images [1, 14, 16].

A number of algorithms have been proposed to produce stereoscopic 3D line drawings from meshes. Most notably, Kim et al. presented a method that produces 3D line drawings by generating contours for left and right eyes separately [14]. Contours are then pruned for view consistency by checking the visibility of points along the curve formed by creating an epipolar plane between a pair of points on the left and right contours. Kim et al. also describe a method for consistent stylization of lines by linking control points between matching contours and applying stylization to the linked pairs. However, this method can only be used with full 3D models. Bukenberger et al. also produce stereo-consistent line drawings from 3D surfaces in object space [4]. Another paper by Kim et al. describes a method for producing stylized S3D line drawings from S3D photographs [15]. Their paper applies Canny edge detection [5] to the edge tangent field [13] of the left stereo image and warps the discovered contours to the right image using the disparity map. However, the rendered lines are from all contours that can be found in the actual image, not only object contours but also texture or lighting contours. By contrast, a hand-drawn stereoscopic 3D line drawing would likely include only object contours and creases. As their method is based on edge detection purely in the colour domain, they cannot differentiate between geometric discontinuities and colour discontinuities. Disparity maps, which indicate the horizontal separation between pixels of the left and right image, isolate geometric information from colour information. Therefore, applying edge detection to the disparity map could uniquely produce object contours. Our method will harness the information in the disparity map to construct S3D line drawings.

## 2.1 Perception and Monoscopic Depth Cues

Our perception of depth arises from both monoscopic (2D) and stereoscopic (3D) depth cues. Monoscopic depth cues include shading, relative size, occlusion, and motion [1, 3]. Shading an S3D line drawing can improve depth perception, but the amount of improvement is limited for images with rich detail [16]. However, for S3D line drawings of simple objects with few internal lines, shading provides the necessary information about object shape. For example, imagine a circular contour: is this a line drawing of a circle or a sphere?

Stereo-consistent shading is complicated by the fact that shading can be view-dependent. Apart from purely Lambertian surfaces, shading features such as specular highlights may be visible in only one eye due to the position of the eyes with respect to the light source and object [2]. This phenomenon can also be observed in S3D photographs, as demonstrated in Figure 2. Note that both specular highlights and reflections differ between left and right views and are circled in cyan for visibility. While these specular highlights are natural to the human visual system, they can be problematic for computer vision algorithms commonly used with stereo [2]. Additionally, it is believed in film that specular highlights can cause binocular rivalry if they are rendered inconsistently between views. Therefore, they are often removed or redrawn to be consistent [17]. We provide users the option of adding shading to our S3D line drawings, to improve the perception of shape. However, our shading will remain true-to-nature. That is, we will not remove or adjust the natural lighting of the S3D images to ensure consistency.

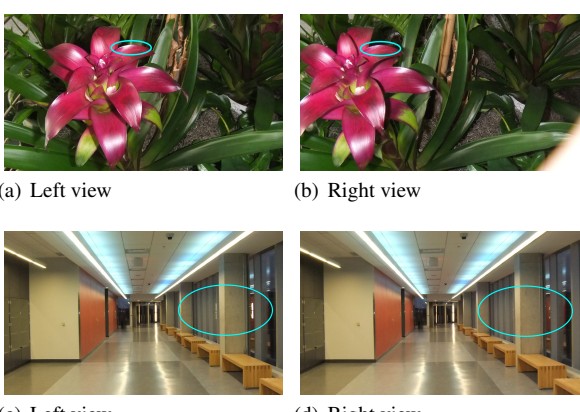

(a) Left view      (b) Right view

(c) Left view      (d) Right view

Figure 2: Inconsistent specular highlights and reflections.

## 2.2 Stylization

Stylization can be applied to both the S3D lines and shaded regions. Surprisingly, it has been shown that simple line styles such as overdrawn lines, varying thickness, and jitter do not negatively impact depth perception and comfort in S3D images if rendered consistently [16].

The naive approach to create consistent stylized lines is to render them in the left view, then use the disparity map and warp (horizontally shift) them to the right. However, the rendered, stylized object contours may have pixels that bleed over onto other surfaces. Therefore, warping individual pixels would not produce smooth lines. Alternatively, the control points of curves or the endpoints of line segments could be warped. But if any of these points from the stylized lines lie on other objects, the lines rendered in the right view may be discontinuous or distorted. Instead, we will match the original control points to their underlying disparities prior to stylization or rendering, similar to the approach used by Kim et al. [14]. Although there are many ways to stylize lines, we focus on overdrawn and jittered styles.

In addition to stylized contours, we will also stylize shaded regions. Stereoscopic 3D image stylization has been well studied, although existing methods focus on stylizing the whole image consistently instead of a small region. Stavrakis et al. applied stylization to the left image and used the disparity map to warp it to the right, then did the reverse for occluded regions [26, 27]. As discussed previously, Northam et al. applied stylization using disparity layers [19, 20]. However, since lighting and specular highlights are view-dependent, these methods would enforce consistency where none exists. Hence, we will apply stylization algorithms to shaded regions in a view-dependent manner to preserve these inconsistencies. By preserving inconsistencies, we contradict Richardt et al. [22] and Northam et al. [19, 20], which focus on establishing or maintaining consistency at all cost. We believe that because shading is a monocular depth cue, binocular rivalry and randomness will have minimal effects on viewer perception.

## 3 PRODUCING A STEREOSCOPIC 3D LINE DRAWING

Our method is divided into several stages and requires left and right images and disparity maps as input. First, the object-depiction contours are extracted. Next, the contours are split into curves by view visibility: left-only, right-only, and shared. Curves are stylized, then warped from left-to-right. Finally, shading may be added to improve depth perception.

### 3.1 Extracting Contours

The shape of an object is described by its silhouette (contours) and interior creases and folds [9, 11]. Both contours and interior lines are needed to give a clear sense of shape [6]. Note that this paper is not concerned with edge detection, but rather with finding object silhouettes, along with interior creases and folds, which we will refer to as contours. While these lines may be found in the S3D image, they are more easily isolated using information in the disparity map.

One possible approach to find contours is to apply edge detection methods to the raw or preprocessed disparity map. Another approach is to perform a 3D reconstruction of the scene using the provided disparity maps and apply a silhouette finding algorithm, such the one presented by Hertzmann and Zorin, to identify the object silhouettes, creases and folds [11]. Yet another approach is presented by Kalnins et al., which focuses on 'frame-to-frame coherence' for animated scenes [12]. Rusinkiewicz et al. also examine several different algorithms for generating line drawings from 3D models [23].

We use the first method, applying edge detection methods to the raw or preprocessed disparity map, instead of discovering silhouettes from a 3D reconstruction. While many of the object silhouettes, creases and folds can be found from the reconstruction using Hertzmann and Zorin's approach, more subtle object contours occurring where two objects intersect at the same depth, along with subtle creases and folds, are not always identified, as shown in Figure 3 [11].

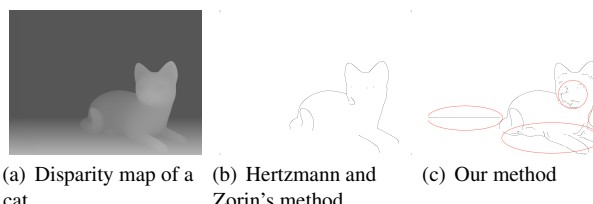

(a) Disparity map of a cat  (b) Hertzmann and Zorin's method  (c) Our method

Figure 3: Hertzmann and Zorin's method vs our method.

After identifying the silhouette contours from the 3D reconstruction, the visibility of those contours must then be computed for each eye, as in Kim et al. [14]. However, visibility is given in the disparity map, so recomputing this information is a waste. Finally, we do not assume that the baseline and focal length of the image is known or can be estimated such that a believable 3D reconstruction can be produced.

#### 3.1.1 Finding Contours in a Disparity Map

There are two types of contours in a disparity map. The first type occurs at a depth discontinuity, where one object occludes another, creating a jump in neighbouring disparity values. These are typically object silhouettes. The second type occurs where two surfaces meet at the same depth, or as creases and folds on an object's surface. The first type can be found using a Laplacian or Canny edge detector, as shown in Figure 4.

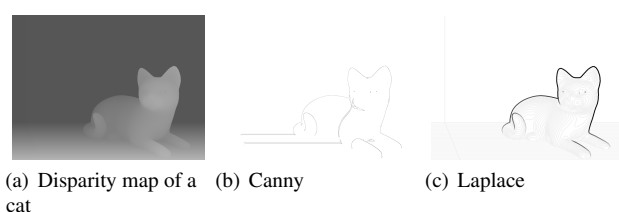

(a) Disparity map of a cat  (b) Canny  (c) Laplace

Figure 4: The strong contours of a disparity map may be found by a Canny edge detector or Laplacian edge detector but they are not ideal.

The second type of contour, created by surfaces meeting at the same depth, or by creases and folds, is more elusive. Adjusting the parameters of a Laplacian or Canny detector can find these contours, but not uniquely, as shown in Figure 5.

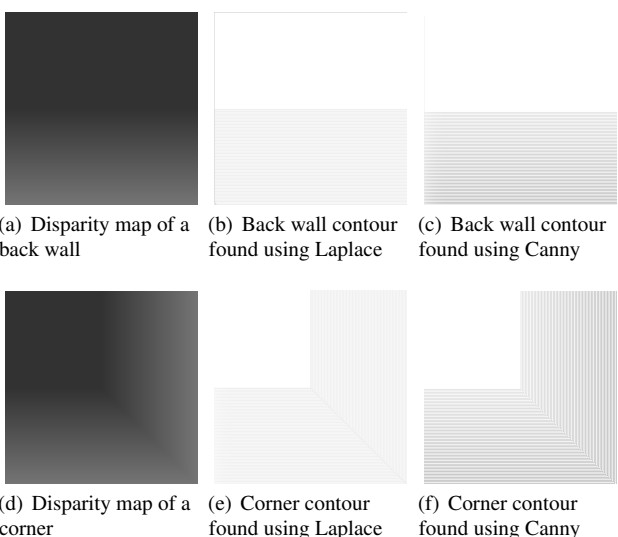

(a) Disparity map of a back wall  (b) Back wall contour found using Laplace  (c) Back wall contour found using Canny

(d) Disparity map of a corner  (e) Corner contour found using Laplace  (f) Corner contour found using Canny

Figure 5: The second type of contour is hard to detect uniquely using Canny or Laplace.

Our method does manage to successfully and uniquely identify these contours, as shown in Figure 6.

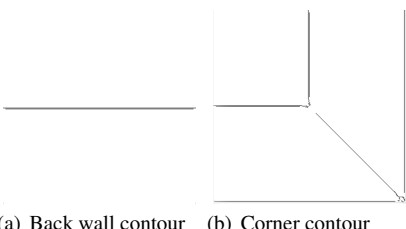

(a) Back wall contour found using our method  (b) Corner contour found using our method.

Figure 6: The second type of contour is easy to detect uniquely using our method

To make these low-contrast contours more visible, we can convert disparity to depth and apply a bilateral filter to smooth the plateaus in the result, as suggested by [18]. However, while this improves the visibility of type one contours, it does not improve visibility for the subtle type two contours that we seek, as shown in Figure 7.

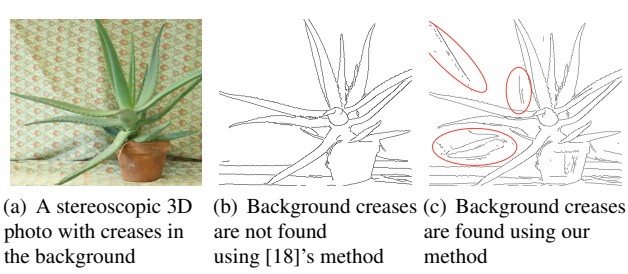

(a) A stereoscopic 3D photo with creases in the background  (b) Background creases are not found using [18]'s method  (c) Background creases are found using our method

Figure 7: Our method improves visibility for type two contours.

We note that finding subtle type two contours, creases and folds in low-contrast regions is known to be a difficult problem [11, 21].

Traditional edge detection algorithms, such as the Laplacian and Canny detectors, are not able to uniquely identify edges in low-contrast areas, as indicated by Savant [24]. And while second-order derivative methods can identify some contours that are zero-crossings, not all type two contours are zero-crossings.

Hence, we propose the following method to identify contours in disparity maps. First, we use a Canny edge detector, as suggested by Gelautz and Markovic, to identify type one contours: the visible object silhouettes [7].

Next, we improve the visibility of type two contours. Hertzmann suggested rendering a scene where different coloured directional lights are cast along positive and negative axes onto a 3D model to produce a brightly-coloured normal map from which object silhouettes, creases and folds could be found [11]. In order to apply this technique to our disparity maps, each pixel needs a surface normal. We assign surface normals by applying a simple surface triangulation to each map. Each pixel position $(x, y)$ is a vertex with depth $z$ equal to the disparity at that position. Eight triangular "faces" are formed by a point $p$ in the disparity map and two of its immediate neighbours. A normal can then be calculated for each of these faces, as well as the vertex normal from the average of the eight adjacent triangular face normals. Thus, we compute a set of triangles from $p$'s 8-connected neighbours and average their normals to find the pixel's normal.

The normals are multiplied by a directional light vector to enhance visualization, as illustrated in Figure 8. However, when directional lights are cast onto the lit normal map, we do not observe a smoothly shaded result as expected. Instead, the normal map appears stepped, with rings of front-facing planes depicted in dark red. This stepped appearance is a consequence of the limited dynamic range of most disparity maps. A perfectly smooth surface cannot be depicted in this discrete space, resulting in many pixels being assigned the same integer disparity instead of their actual values. These artificial contours make discovery of actual contours, such as the interface between the wall and floor, difficult to achieve.

In order to remove the stepped or plateaued appearance, floating point disparities are needed, along with a smoothing operator to reduce the discretized appearance. Ideally, converting the disparity map from 8-bit to floating point and applying a simple out-of-the-box smoothing operation would smooth these plateaus out. However, directly applying a bilateral filter will preserve or enhance these contours and a Gaussian or box filter would soften all contours indiscriminately, effectively blending objects together. Instead, to smooth these plateaus and generate a smooth lit disparity with preserved contours, we:

1. Compute the strong contours using a Canny filter, dilating the result to produce a contour mask where contours have a diameter of 10 pixels.

2. Calculate the surface normals via triangulation as previously discussed. Use larger triangles (8 pixels in height) in regions that do not correspond to contours in the aforementioned mask and smaller triangles (1 pixel in height) along contours. This hybrid approach gives us clear, prominent lines corresponding to key contours, and additional smoothing elsewhere. Do this for both the discrete (un-smoothed) and floating point (smoothed) disparity maps.

3. Cast directional lights to colourize and produce the smoothed and un-smoothed maps. This yields Figure 8(a) and Figure 8(b), respectively.

4. Compute the complexity of the discrete (un-smoothed) disparity map, $\alpha$, as the number of observed disparities. Apply a bilateral filter to the smoothed map $\frac{\alpha}{10}$ times. Note that $\frac{\alpha}{10}$, and other parameter values, were selected after applying the method to our test set of 12 images and selecting the parameter value that produced the best results overall for all images.

5. To correct the blown-out contours caused by the previous step, extract the strong mask contours from the un-smoothed map (that is, the pixels of the un-smoothed map coinciding with the pixels of the mask generated in step 1) and superimpose them on the smoothed map. Apply a bilateral filter to the smoothed map $\frac{\alpha}{10}$ times to help blend the contours in.

6. Overlay the original, un-dilated version of the mask on top of the smoothed map, as seen in Figure 8(c), to aid in the identification of strong contours.

We can now apply the Canny edge operator to the smoothed and coloured map in an automated fashion.

In general, we seek to compensate for less detailed masks with more permissive Canny thresholds that yield a more detailed final contour set. Conversely, more detailed masks imply that a stricter threshold should be used, to prevent an overly noisy final contour set. Let the number of mask pixels be $\beta$. Let $\beta$ divided by the total number of pixels be $x$. We recognize that the level of detail in the mask, $x$, is inversely proportional to the number of pixels desired in the final contour set.

We also want to take into account the aforementioned disparity map complexity, $\alpha$. This is another inversely proportional relationship, between disparity map complexity and the target number of pixels in the final contour set. The more complex the disparity map is, the greater the number of easily identifiable contours, and the less permissive the Canny threshold is required to be. Conversely, the less detailed the disparity map, the more difficulty Canny will have extracting contours from it, and the more permissive we should make the Canny threshold.

We will use these two complexity measures – and the direct, inversely proportional relationships that we observed – to select Canny thresholds automatically. In so doing, we are letting each image speak for itself and lifting the burden of fine-tuning from the user.

Let $\phi = \frac{1}{\alpha x}$ be the target number of pixels in the final contour set.

We want the final contour set to contain at least as many pixels as the mask; only then can more contours be found to supplement the mask contours. Therefore, we modify our target to $(1 + \sigma)\beta$ where $\sigma = min(3, \phi)$. Notice that we set a cap of 3 on $\phi$. This is because, experimentally, we have found that larger values introduce a lot of noise.

What we have established is a target number of final contour pixels, not a Canny threshold parameter. But each Canny threshold parameter will produce a certain number of contour pixels. The higher the threshold, the less pixels in the final result; the lower the threshold, the more pixels in the final result. The minimum threshold parameter is `min=0` and the maximum is `max=255`. Using these boundaries, we can conduct a binary search for the ideal parameter. We start by using a threshold parameter midway between `min` and `max`. We then count the number of pixels in the resulting contour set. If the result is below target, we need to be more permissive, so we lower our `max` and try a lower threshold in the next iteration. If the result exceeds the target, we need to be less permissive, so we increase our `min`. We stop once `max=min`, or the contour pixel count equals the target.

Once contours are found, we use the `findContours` function in OpenCV to extract curve points from the raster image. The extracted curve points are processed to remove curve duplication. The curves are also split whenever adjacent point disparities differ by more than a small threshold, under the assumption that the adjacent points belong to separate surfaces. We note that some of the original detail may be lost in this process.

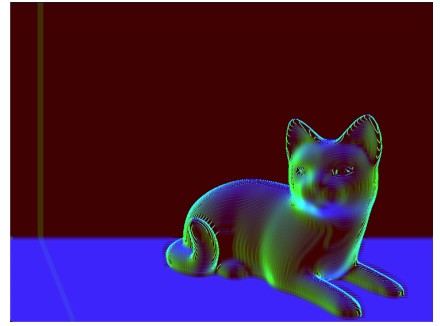
(a) Raw disparity map directionally lit

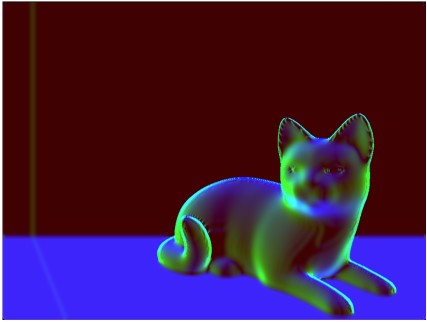
(b) Smoothed disparity map directionally lit

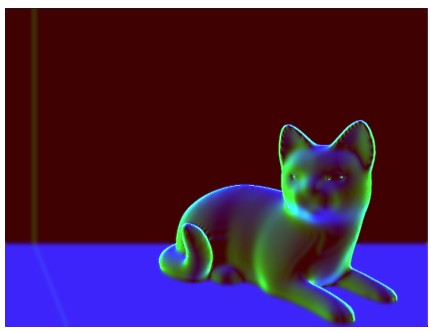
(c) Smoothed disparity map directionally lit after processing

Figure 8: Directionally lit disparity maps.

### 3.1.2 Splitting Lines for Consistency

As aforementioned, OpenCV's `findContours` function produces a set of control points, which are used as inputs to the ImageMagick Bezier rendering function, `DrawableBezier`. Contours extracted from the disparity maps are thereby rendered as smooth curves for the corresponding view. However, view dependent rendering introduces inconsistencies, which arise when contours are found in one disparity map but not the other due to the slight variation in the viewpoint.

To prevent these inconsistencies – which cause discomfort – we arbitrarily select the left view to be the "true" contours. Then, any line in the left view that is visible in the right is warped using the disparity map to the right view. Lines only visible in the left are rendered only in the left; likewise, lines only visible in the right are rendered only in the right. View visibility is determined by the disparity map. Any pixel $p(x, y)$ is visible in both the left and right views if $L(x, y) = d$ and the corresponding pixel $R(x - d, y) = d$, where $L$ and $R$ are the left and right disparity maps respectively, and $d$ is the disparity of pixel $p$.

Contours are warped by their control points and then rendered in a view-dependent manner. While this method of rendering potentially introduces inconsistencies, warping the rendered lines would introduce noise as the lines may lie on surfaces with different disparities.

Finally, we note that long contours extracted from the disparity map may span multiple objects and both occluded and visible regions. Warping the entire stroke can result in partially occluded contours being visible in the wrong view. To prevent this, we split strokes whenever the visibilities of adjacent control points change, i.e. from visible to occluded. We also split strokes when the disparities of adjacent control points differ by more than some threshold $\tau_d$. Note that we used $\tau_d = 5$, as we observed that curve points are close together, with no large jumps in depth. Strokes that cannot be warped, because they are only visible in one view, are rendered in only one view.

### 3.1.3 Consistent Control Point Stylization

Monoscopic and S3D line drawings are often stylized and represent objects using rough, overdrawn and jittery lines. To increase visual interest, we provide the option of stylizing S3D lines with an overdrawn or jittered style.

Kim et al. discussed a method for stylizing stereoscopic 3D lines [14]. Their method performs stylization after lines have been discovered for both left and right images. Specifically, it links line segments in the left view to the matching segments in the right and consistently renders texture to these linked and parameterized curves.

Our approach is similar and stylizes lines prior to warping by replicating and transforming control points. To produce overdrawn lines, curves are duplicated a fixed number of times. Lines can then be scaled (about their centroids or the centre of the image) by a small random factor, so that the overdrawn lines are visibly distinct. A jittered or rough appearance is created by adding small random translation vectors to each control point of a line. Note that, prior to altering the control points of a line, it is important to store the original, pre-transformed disparities of those control points, so that they can be correctly warped after stylization.

## 4  SHADING

S3D line drawings depict the shapes of objects. These lines do not convey information such as surface texture or roundness, but shading and highlights do. Object shading and shadows are monocular depth cues [8]. Shading, particularly involving specular highlights, is view dependent [28]. Adding monocular depth cues to S3D line drawings can improve the viewer's understanding of surface shape and enhance depth perception. Also, because lighting is view dependent, the left and right views will be stylized independently to preserve their separate lighting characteristics.

To produce the stylized shading, the left and right input images are converted to grayscale and stylized using a variety of algorithms. While any stylization algorithm or filter could be used, we chose those that do not explicitly render contours, as our method will produce those separately. Finally, the stylized shading and S3D lines are combined to produce the final image.

## 5  RESULTS

We tested our method on several S3D images, some of which are shown in Figure 9. Seventy-five percent of the images used as input to our method have high-quality or near-perfect disparity maps.

Figure 10 illustrates some of the 3D line drawings produced by our method. Note that, since lines are generated from the disparity map, contours and interior lines are the only lines visible.

Stylizing the S3D lines yielded the images depicted in Figure 11. Note that even with jittered and overdrawn lines, the left and right views remain consistent.

We used three types of stylization for shading: toon-like shading produced by quantization of the RGB image, impressionist, and halftoning with large particles. None of these stylizations explicitly render contours, so there is no overlap between shading and the line drawing. We combined the stylized S3D line drawings with stylized shading to produce our final images, some of which are shown in Figure 12. To ensure the visibility of the S3D lines, we reduced the darkness of shaded regions by 50%.

We also applied our method to S3D photos with computed disparity maps. These maps contain noise, disparity mismatches, and obscured object contours, which pose a challenge for many S3D algorithms. Figure 13 demonstrates our method's performance on a

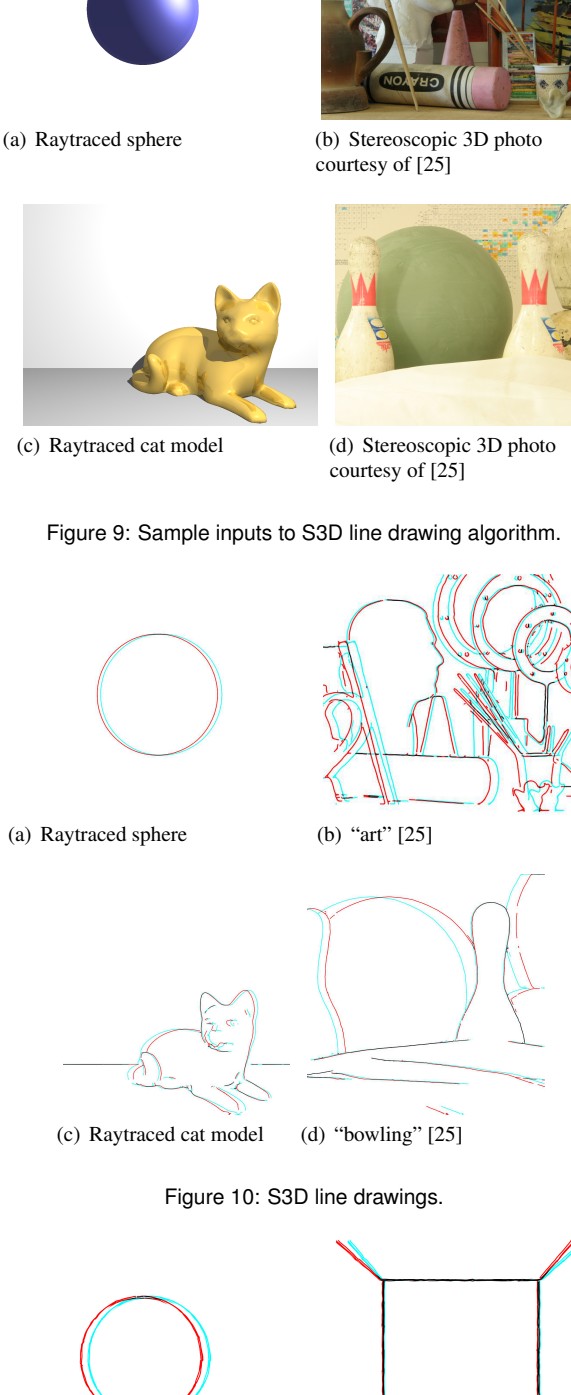

(a) Raytraced sphere

(b) Stereoscopic 3D photo courtesy of [25]

(c) Raytraced cat model

(d) Stereoscopic 3D photo courtesy of [25]

Figure 9: Sample inputs to S3D line drawing algorithm.

(a) Raytraced sphere

(b) "art" [25]

(c) Raytraced cat model

(d) "bowling" [25]

Figure 10: S3D line drawings.

(a) Jittered lines for a rough appearance

(b) Tapered lines for a thick pen appearance

Figure 11: S3D line drawings (red/cyan anaglyph).

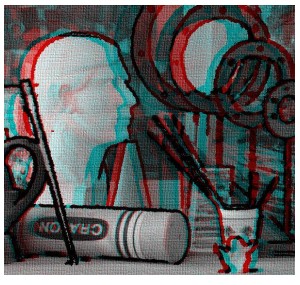 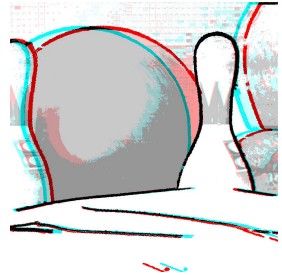

(a) GIMPressionist styled shading  (b) Toon shading

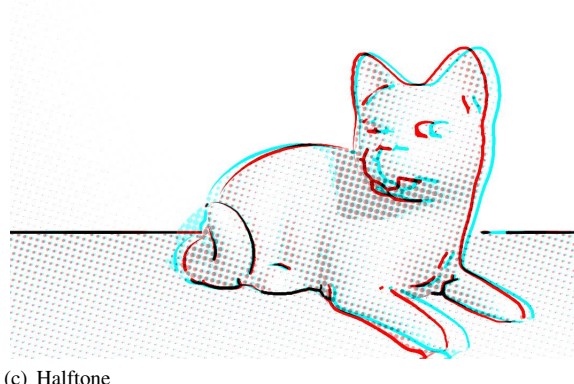

(c) Halftone

Figure 12: Final stylized S3D line drawings (red/cyan anaglyph).

S3D photo with a low-quality disparity map. Despite these disparity errors, our method is still able to produce line drawings, as demonstrated in Figure 13. Note how some lines appear to be missing, typically because they are not visible in the disparity map.

## 6 EVALUATION

To evaluate our results for quality of depth reproduction and viewing comfort, we conducted a short study. For health and safety reasons due to COVID-19, our study was conducted remotely. We asked participants to view a set of 24 images from our dataset using either anaglyph glasses, a 3D TV, a VR headset, or by free-viewing in their homes. For each image, participants were asked to rate the viewing comfort and apparent depth on a Likert scale from 1 to 5. Participants were also asked to rate how aesthetically pleasing they found each image. Images were randomized, and participants were not aware of what they would be viewing.

Overall, participants found that our consistent line drawings were more comfortable, reproduced a greater sense of depth, and were more aesthetically appealing than the raw, inconsistently-rendered line drawings. Table 1 indicates the average difference between each of our results and the inconsistently-rendered line drawings. This difference is a percentage increase from the raw, unstylized lines to our method. So, for example, the first cell demonstrates that the average score was 26% better for our consistently-rendered, unstylized lines than for raw, unstylized lines rendered inconsistently. Note that adding stylization to our lines improved comfort, depth reproduction, and overall aesthetic. We expected participants to find the stylized lines more aesthetically pleasing, but we did not anticipate they would find these more comfortable or conducive to a greater sense of depth. This may be because the stylized lines are more prominent than the unstylized lines, providing participants

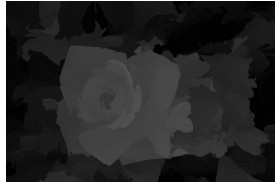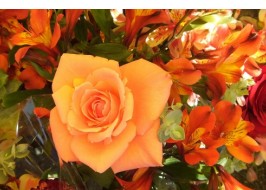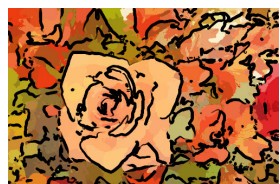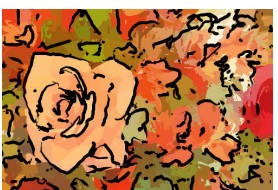

(a) Low resolution disparity map, left view    (b) Original image with low resolution disparity map, left view    (c) Stylized line drawing with shading, left view    (d) Stylized line drawing with shading, right view

Figure 13: Line drawings from a photo with a low-quality disparity map.

with more visual information to fuse and resulting in greater viewing comfort and depth. Also note that adding shading, a monocular depth cue, significantly improved all metrics, regardless of how that shading was rendered. Even the halftone/newsprint shader applied inconsistently, which renders large circles into the scene, was more comfortable, produced more depth, and was more aesthetically pleasing than plain lines. This is interesting, because these images were 55% less consistent than our plain line drawings. We computed consistency by comparing the colour values of pixels that should match according to the disparity map.

Table 1: The difference in comfort, depth reproduction, and aesthetic appearance between raw, unstylized and inconsistently rendered lines and our method. Note that the averaged participant scores for view-dependent, unstylized lines are presented in Table 2.

|  | comfort | depth | appearance |
|---|---|---|---|
| our unstylized lines | 26% | 14% | 20% |
| our stylized lines | 32% | 16% | 26% |
| our unstylized lines with consistent shading | 46% | 46% | 71% |
| our unstylized lines with inconsistent shading | 27% | 42% | 41% |
| our stylized lines with consistent shading | 48% | 45% | 59% |
| our stylized lines with inconsistent shading | 52% | 41% | 66% |

Table 2: Averaged participant scores for view-dependent, unstylized, and inconsistently rendered lines that were used to to compare our various methods to. Note that we have provided averaged participant scores for stylized lines with consistent shading for reference.

|  | comfort | depth | appearance |
|---|---|---|---|
| view dependent, unstylized lines | 2.5 | 2.6 | 2.1 |
| our stylized lines with consistent shading | 3.7 | 3.8 | 3.4 |

Ideally, our participants would be a random sample of individuals with varying backgrounds and exposure to S3D. However, as we were required to run this study remotely, we relied on finding individuals that owned their own S3D viewing equipment or were able to free-view. Hence, our participant pool was drawn from individuals that could be considered S3D enthusiasts. Consequently, participants were critical, and quick to identify and articulate flaws in images, such as window violations and ghosting. Nevertheless, we appreciated their honest and experienced assessments as they provided a clearer and more concise evaluation of our results.

We also note that the study conditions were not ideal. Firstly, we relied on participants to self-report their ability to perceive depth. Secondly, due to the rarity and variety of S3D viewing equipment available, it is unlikely that any two participants used the exact same viewing technology. We categorized viewing mechanisms into three groups: anaglyph, 3DTV/3DS/VR, and free-viewing. Of the 16 participants, 50% used anaglyph glasses, which are prone to crosstalk and ghosting that may cause discomfort. A smaller number of participants, 31.2%, used some other 3D viewing apparatus, such as a 3DTV. This technology may exhibit some crosstalk or ghosting, but significantly less than anaglyph glasses, typically making this technology more comfortable to use. Finally, about 18.8% of par-

ticipants free-viewed the images. The study conditions may have thereby contributed disproportionally to viewing discomfort.

## 7 CONCLUSION AND LIMITATIONS

Our algorithm successfully produces stylized stereoscopic 3D line drawings from photographs. These line drawings reproduce 3D shape, especially when combined with monoscopic shading. Furthermore, for fine-grained stylizations, inconsistent shading did not have a negative impact on the perception of depth or comfort. As expected, large-grained stylizations, such as halftoning, were not as comfortable as their consistently-shaded variations.

A major limitation of our method is that the quality of our method's results largely depends on the quality of the disparity maps provided. Noisy, non-smooth disparity maps, as well as those with obfuscated or obscured object contours, will likely produce noisy line drawings where the object contours are not clearly visible. This, in turn, may produce line drawings with no identifiable subject. Overcoming this limitation is the subject of future work.

Moreover, some parameter selections in Section 3.1.1 – such as the empirical choice of $\frac{\alpha}{10}$ after limited experimentation with 12 images, clamping $\phi$ to a maximum of 3, and dilating contours to 10 pixels in the contour mask – were chosen using our input dataset. Further analysis with a larger dataset could yield more appropriate values in general.

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
