# OpenReview forum: "Generating Rough Stereoscopic 3D Line Drawings from 3D Images"
_graphicsinterface.org/Graphics_Interface/2021/Conference/Second_Cycle — GI 2021_

### Official Review · Reviewer_rAxd · 2021-04-30
**missing detail and muddled explanation undermine the paper's method**

**Rating:** 5
**Confidence:** 4

**Review:**

This is the second time I have reviewed this paper. In my earlier review, I raised two main concerns: first, that the organization was somewhat lacking and background on S3D was missing; second, that the problem of edge detection in disparity mapping did not justify the presented method. The first complaint has been addressed but the second problem might be even worse in this version. I am mildly recommending rejection, but I do not have strong feelings either way. I do think that the paper should not be published as is, but it may be possible to repair the presentation in the revisions phase.

The organization has generally been improved. The opening paragraph provides some context on S3D and overall the standard of writing has risen, though I do question the use of footnotes to introduce definitions (the definitions should be in the main text).

The discussion of edge detection should still be redone, though. The description is clearer, and I can now see that the basic issue is that "edge detection" is not the desired concept at all. As stated at the beginning of 3.1, the desired lines are object silhouettes plus internal creases and folds, i.e., geometry-space concepts. Image edges are discontinuities in color or intensity; changes in the derivative (with c1 continuity) are not edges. Figure 5 shows an attempt to recover discontinuities in the normal by searching for an intensity discontinuity; that is bound to fail, and referring to the entity as an "edge" confuses the issue.  Similarly, the lower boundary between the cat's leg and the floor in Figure 3 is not an edge in the image sense (nor is it a silhouette in your recovered geometry). More care with the terminology would have prevented some wasted effort.

"We note that finding type two edges, or finding edges in a low-contrast or noisy region, is known to be a difficult problem [19]." But your "type two edges" are not edges at all. This has nothing to do with contrast or noise. This discussion is highly misleading. You cite Cole et al. on where people draw lines; it would be worth tracing back this line of work to papers like DeCarlo et al.'s suggestive contours and Judd et al.'s apparent ridges.

The basic idea for finding the features of interest (creating normals and then applying different lights to make normal discontinuities stand out) is fine, if not new to this work. I had a hard time following the description of how normals were obtained, though. Some details are not given and some that are given are confused.

"Triangular “faces” are formed by a point p in the disparity map and two of its immediate neighbours, q and r." Which two neighbours? I suppose you will do this for all four combinations of 4-connected neighours. But then: "A normal can then be calculated for each of these faces, as well as the vertex normal from the average of the eight adjacent triangular face normals."  What are the eight adjacent triangles, and adjacent to what? If you mean that you are computing a set of triangles from your 8-connected neighbours and averaging those to find the pixel's normal, surely that could be stated more clearly. I am a bit perplexed about this process, though, since it yields different triangles for adjacent pixels (a quad's diagonal is
split differently depending on which corner you are considering).

The paper states that larger triangles are used depending on the distance from an edge. How this works is not stated. Are the edges in question the initial Canny edges (i.e., actual discontinuities?) If you are just taking two or three steps to obtain the vertices to link into triangles, this seems strictly worse than filtering. If you are doing something more sophisticated, describe it.

In my previous review, I dismissed the edge detection problem. I can see that there is a real problem there (albeit not one of edge
detection, since the features of interest are generally not edges). I am still not convinced by the solution which still seems ad hoc, perhaps even moreso with the added details or complications beyond what was presented in the earlier paper.

Should this paper be accepted, I recommend rewriting section 3 to to more accurately describe the problem, removing figures 4 and 5 (and possibly 3), and adding the missing details and clarifications to the algorithm description.

Minor comments:

I still think that reporting scores (like Table 2) and not percentages (Table 1) gives a clearer picture of the data.

"There are two possible approaches to find contours (edges)." First, equating contours and edges is a mistake, as discussed above. Second, while you describe two possible approaches, there are probably an infinite number, depending on what you count as being a different approach.

"...a silhouette finding algorithm, such as Hertzmann and Zorin" Hertzmann and Zorin are people, not an algorithm. Write something like "such as that of Hertzmann and Zorin." Similar conflations of authors and their work appear in other places in the paper. Fix these too.

"Apply a bilateral filter alpha^3/10 times." Don't mix footnotes and symbols, as the footnote looks like a mathematical operation.

You report participants to three digits of accuracy (e.g., "31.2% of participants") when you typically mean a single-digit count of your 16 participants. It would be better to report a fraction (e.g., 5/16). Along similar lines, the wording "about 18.8% of participants" provokes confusion by combining a statement of uncertainty with a highly specific number. (Ought we to imagine that the actual number is between 18.7% and 18.9%? Likely not.)

"In so doing, we are letting the image – not the user – do the talking." Why silence your user? I am not sure what to make of this sentence.

"Directly applying a bilateral filter will preserve or enhance these edges." Enhance, really? Depending on the settings, it might not reduce the quantization much, but I have a hard time seeing how the quantization artifacts would become larger.

---

### Official Review · Reviewer_sTmz · 2021-05-04
**The submission is a sufficiently detailed and well-written novel stereo 3D line stylization edge extraction contribution alongside a limited perception study. The paper should merit publication given the overall quality, novelty of the approach and the new image consistency findings, which are contrary to closely related prior work.**

**Rating:** 7
**Confidence:** 3

**Review:**

The work makes use of Stereo 3D images and disparity maps in an original way to produce stylized line drawings with optional stylized shading. The authors clearly distinguish how this work differs from relevant prior work while also being sincere about limitations.  Results are provided from a relatively limited perception study, involving 16 remote participants, and a small collection of processed images is shared in the supplementary materials for additional evaluation.
The manuscript is well-written, clear and flows smoothly.  There is some novelty in the new line stylization approach to achieve better edge extraction for inner edges with a well-documented procedure.  This work should be of some significance to the NPR/stylization communities, given some additional findings regarding preserving consistency compared to prior work, and the recommendation is to accept the submission for publication.

Comments:
* The empirical choice of $\alpha / 10$ after limited experimentation with 12 images can be more thoroughly assessed.
* Some additional justification for clamping $\phi$ to max of 3, beyond the mention of excessive noise, might be desirable.
* It is unclear what and whether there are any significant implications for the arbitrary choice of a 10 pixel edge dilation.
* A shortcoming of the approach is the need for relatively high quality disparity maps as input, while tackling this issue is left as future work, it could be worthwhile expanding a bit on it in the current work.
* Section 4 covering shading is rather short relative to the rest of the paper. Does it merit its own separate section and, if so, should it be expanded somewhat in this work or could it just be included elsewhere?
* Window violations and ghosting artifacts are acknowledged but could also benefit from some additional discussion.
* The submission identifies perception study findings that are somewhat contrary to the findings of prior work with respect to inconsistent shading. Perhaps this can be further highlighted in the abstract and conclusion.
* The supplementary material could benefit from some additional documentation and viewing recommendations. Will the pipeline codebase and a larger image dataset be shared with the community?

Some other minor corrections/improvements for the manuscript:
* Figure 4: Correct the spacing in the caption " may b efound" -> "be found"
* In Section 3.1.1: "a smoothly shaded cat as expected" Consider adding some more context to this sentence as mention of the "cat" figure appears in the text and only makes sense when we look at the actual figures.
* Footnotes: The symbols for some of the footnotes (3;4) can be a little confusing in the text and potentially could initially be mistaken for a power term. Consider changing the annotation to make it more clear.
* Figure 13: Consider increasing the horizontal gap between subfigures as the captions of (b) and (c) are blending together.

---

### Official Review · Reviewer_DizZ · 2021-05-04
**Better than the previous iteration, some comments left unaddressed**

**Rating:** 6
**Confidence:** 3

**Review:**

The paper presents a new method for stereoscopic contour generation from a stereoscopic image with its disparity map.  Starting with a simplistic 3D reconstruction of the scene, they render it and extract contours via an image-space contour extraction algorithm (Canny). Doing so for two viewpoints, they resolve the inconsistencies by warping left contours to the right, splitting long lines/lines spanning multiple objects, using the disparity map. Finally, they stylize the contours and add shading.

I reviewed a previous submission of this paper some time ago, and I am slightly confused by this one. As I indicated in the previous review, the paper would benefit from these references, which seem very relevant, _especially_ the last one.  The last one, in my view, suggests that since the proposed method produces some sort of 3D reconstruction, then perhaps [3] would just provide the contours they're looking from automatically.

[1] Robert D. Kalnins, Philip L. Davidson, Lee Markosian, and Adam Finkelstein. 2003. Coherent stylized silhouettes. In ACM SIGGRAPH 2003 Papers (SIGGRAPH '03)
[2] Szymon Rusinkiewicz, Forrester Cole, Doug DeCarlo, and Adam Finkelstein. 2008. Line drawings from 3D models. In ACM SIGGRAPH 2008 classes(SIGGRAPH '08).
[3] Stereo-consistent Contours in Object Space by Dennis R. Bukenberger, Katharina Schwarz and Hendrik P. A. Lensch

Also, while some of my comments got addressed, this comment did not, and I'm again unclear why.
The text (3.1.2) suddenly refers to smooth splines? How are those formed from Canny edges? Seems non-trivial, and there are no details. I'm assuming this is just a careless choice of words, I don’t think there are actual splines. However, later, the authors mention 'control points' multiple times, so I'm not sure what to think. If those are the contour generators (i.e. the curves in 3D that become contours when projected), then the terminology and the explanation should be made clearer. But even if so, those would be polylines, not splines…

However, compared to the previous iteration, the most important addition the authors have made is adding the user study that validates the method. I think this is a very important addition, and the results (especially with inconsistent shading) are very curious, if they are reproduced in some follow-up work.

Having said that, I think the method and the application is interesting. Even though some of the algorithmic choices are not well justified and a few important references are missing, I think with the validation the paper can be published. I would still love the final version of the paper to address the comments above -- or explain to me why I am wrong.

---

### Meta-Review · Area_Chair_zAF7 · 2021-05-05

**Recommendation:** Accept
**Confidence:** 3

**Metareview:**

The reviews are sufficiently positive that this paper can be accepted. However, some changes are required to bring this paper up to standard. Reviewer DizZ suggests some additional references; indeed, there are several SIGGRAPH papers from the early 2000s that propose various mechanisms for finding contours, valleys, and ridges from geometry that this paper would benefit from citing. Reviewer DizZ also raises the question of how the splines are determined, which should be discussed in the paper (perhaps the curves are not really splines at all). Reviewer sTmz has several worthwhile comments that can be addressed. Reviewer rAxd raises concerns about the term "edges" in a way complementary to the review suggestions from DisZ and asks for more accurate terminology use and the removal of some superfluous figures, as well as pointing out some gaps in the algorithm description that should be filled before publication.

The chairs may wish to consider the brief period available for revision. I would not want to see the misinformation presently in the paper enter the literature. However, focussed effort can easily complete an adequate set of revisions in the time available. I suppose there is no mechanism for checking whether the comments have been addressed, though if "shepherding" is an option this year then I am willing to double-check the revised paper.

---

### Decision · Program_Chairs · 2021-05-08

Accept